# Visual Prompt Engineering for Vision Language Models in Radiology

**Stefan Denner**[1,2]                                   STEFAN.DENNER@DKFZ-HEIDELBERG.DE

**Markus Bujotzek**[1,3]                             MARKUS.BUJOTZEK@DKFZ-HEIDELBERG.DE

**Dimitrios Bounias**[1,3]                          DIMITRIOS.BOUNIAS@DKFZ-HEIDELBERG.DE

**David Zimmerer**[1]                                      D.ZIMMERER@DKFZ-HEIDELBERG.DE

**Raphael Stock**[1,2]                                  RAPHAEL.STOCK@DKFZ-HEIDELBERG.DE

**Klaus Maier-Hein**[1,2,3]                             K.MAIER-HEIN@DKFZ-HEIDELBERG.DE

[1] *Division of Medical Image Computing, German Cancer Research Center, Heidelberg, Germany*

[2] *Faculty of Mathematics and Computer Science, Heidelberg University, Heidelberg, Germany*

[3] *Medical Faculty Heidelberg, University of Heidelberg, Heidelberg, Germany*

**Editors:** Accepted for publication at MIDL 2025

## Abstract

Medical image classification plays a crucial role in clinical decision-making, yet most models are constrained to a fixed set of predefined classes, limiting their adaptability to new conditions. Contrastive Language-Image Pretraining (CLIP) offers a promising solution by enabling zero-shot classification through multimodal large-scale pretraining. However, while CLIP effectively captures global image content, radiology requires a more localized focus on specific pathology regions to enhance both interpretability and diagnostic accuracy. To address this, we explore the potential of incorporating visual cues into zero-shot classification, embedding visual markers, such as arrows, bounding boxes, and circles, directly into radiological images to guide model attention. Evaluating across four public chest X-ray datasets, we demonstrate that visual markers improve AUROC by up to 0.185, highlighting their effectiveness in enhancing classification performance. Furthermore, attention map analysis confirms that visual cues help models focus on clinically relevant areas, leading to more interpretable predictions. To support further research, we use public datasets and provide our codebase and preprocessing pipeline here, serving as a reference point for future work on localized classification in medical imaging.

**Keywords:** Vision-Language Models, Localized Classification, Zero-Shot Classification

## 1. Introduction

Medical image classification remains a long-standing and critical problem in the field of healthcare. Despite the advances in automatic classification approaches, these methods are typically limited to the few specific pathologies they were trained on (Holste et al., 2024). This limitation is particularly pronounced due to the vast range of potential pathologies and the insufficient availability of comprehensive training data (Langlotz, 2023). In contrast, model architectures like Contrastive Language-Image Pre-training (CLIP) (Radford et al., 2021), have shown great performance by not training for a specific classification task, but

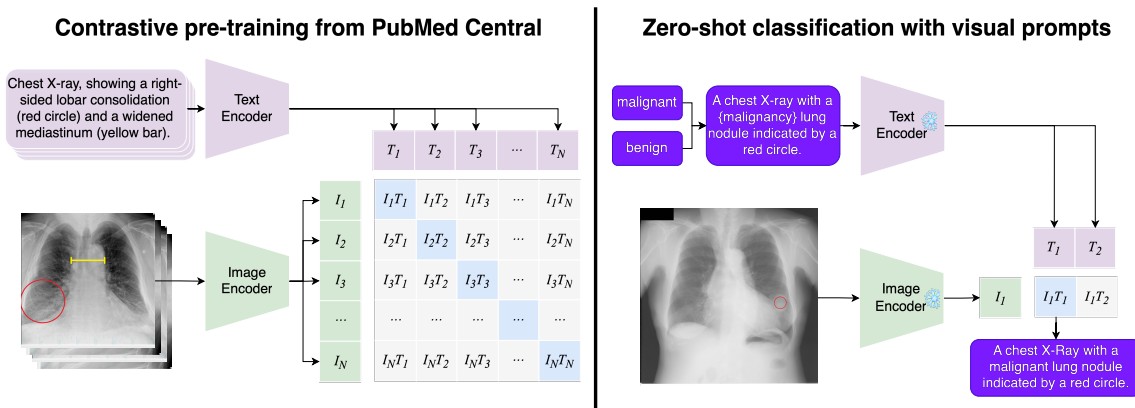

Figure 1: Training paradigm of CLIP (left) and how we use it for zero-shot classification (right). CLIP pretrained on biomedical image-text pairs from scientific articles learns semantical representations aligning image and text. For zero-shot classification, we provide the target image (with a visual marker) and text descriptions of the potential classes. Example image (left) from (Chowdhury et al., 2018).

leveraging the large corpora of text-image pairs for pre-training. CLIP demonstrates that their pre-training task of predicting which caption goes with which image is an efficient and scalable way to learn state-of-the-art image representations. After pre-training, CLIP enables zero-shot classification by leveraging natural language to reference visual concepts (Radford et al., 2021). While CLIP captures global image content, certain applications require a more fine-grained focus on specific regions of interest. This limitation is particularly crucial in radiology, where pathological structures are often small, subtle, and challenging to detect. Moreover, CLIP's global perspective becomes insufficient when multiple regions of interest exist within a single image (Sun et al., 2024), a common scenario in radiology where multiple pathologies frequently coexist in the same scan (Castro et al., 2024; Wang et al., 2017; Nguyen et al., 2022). Additionally, radiologists often identify abnormalities but require assistance in classifying them (Yildirim et al., 2024). This highlights the need for models that can prioritize and interpret localized pathological regions rather than relying solely on global image representations. An intuitive approach to let the CLIP focus on a specific region would be to crop the image. However, this loses the global context of the pathology, therefore might harm classification performance. Recent works in the natural image domain investigated to draw markers directly on the image leading to state-of-the-art results in zero-shot tasks (Shtedritski et al., 2023; Yang et al., 2024). They hypothesize that the model has seen the chosen visual markers during training and understands the meaning behind them. However, they also indicate that this behavior is more likely to be learned from large datasets and high-capacity models, given the scarcity of such visual markers in the training data (Shtedritski et al., 2023).

In radiology, due to limited data availability, a common strategy for training Vision-Language Models(VLMs) involves utilizing public research articles (Eslami et al., 2023; Zhang et al., 2023a,b; Lin et al., 2023; Lozano et al., 2025; Pelka et al., 2018; Subramanian

et al., 2020). Given the prevalence of visual markers in scientific images (see Appendix Fig. 3), we hypothesize that VLMs trained on these datasets, despite being smaller than their natural image counterparts, can still recognize and interpret such markers. This capability may enable them to leverage visual cues to guide attention and influence decision-making. Therefore, this work investigates whether visual prompt engineering, i.e. embedding markers within radiological images, enhances zero-shot classification performance. We evaluate our hypothesis on multiple chest X-ray datasets. Beyond quantitative analysis, we also provide evidence that the model truly recognizes the visual markers by visualizing attention maps. To our knowledge, this is the first study to investigate visual prompt engineering in the radiological domain.

## 2. Methods

### 2.1. Zero-shot Classification with CLIP

CLIP (Radford et al., 2021) classifies images in a zero-shot manner by embedding images and text into a shared space. CLIP consists of two separate encoders: one for images and one for text. Given an image $\mathbf{I} \in \mathbb{R}^{3 \times H \times W}$, CLIP's image encoder produces an embedding $\phi(\mathbf{I})$, while the text encoder maps an input text $t \in \Sigma^*$ to an embedding $\psi(t)$. Both embeddings lie in a shared latent space. A compatibility score $s(\mathbf{I}, t) = \text{cosine}\big(\phi(\mathbf{I}), \psi(t)\big)$ is computed, by using the cosine similarity between the image and text embeddings (Radford et al., 2021).

To perform classification over $N$ candidate classes, we first define a set of text prompts $\{\mathbf{T}_i\}$ for $i \in \{1, 2, \ldots, N\}$. Each $\mathbf{T}_i$ describes a class. We then compute the similarity scores $\{s_i\}$ by evaluating $s_i = s(\mathbf{I}, \mathbf{T}_i)$, for each class $i$. These similarity scores are interpreted as logits for a softmax function:

$$P\big(y = i \mid \mathbf{I}, \{\mathbf{T}_i\}\big) = \frac{\exp\big(s_i\big)}{\sum_{j=1}^{N} \exp\big(s_j\big)}. \tag{1}$$

The final predicted class $\hat{y}$ is taken to be the one with the highest softmax probability:

$$\hat{y} = \arg\max_i \ P\big(y = i \mid \mathbf{I}, \{\mathbf{T}_i\}\big). \tag{2}$$

### 2.2. Visual Prompting

While encoding an image into a global embedding is effective for broad categorization tasks, this global view can overshadow small or subtle findings. This is particularly problematic in radiology, where pathologies are often localized and subtle. Moreover, multiple pathologies may appear simultaneously in a single scan, each requiring targeted attention. Therefore, it is essential to develop approaches that direct VLMs' attention to specific regions of interest, rather than relying solely on global image features. A common approach to incorporate region-specific information into image classification pipelines is cropping, where the image is truncated to the region of interest. This effectively reduces distractions but risks losing global context, which is often critical in radiological assessment. Some works, including (Ma et al., 2024; Kirillov et al., 2023; Sun et al., 2024), integrate region-specific prompting techniques directly into model architectures. These approaches, however, require dedicated

training on task-specific data with precise spatial annotations, which is costly and often infeasible in medical imaging. In contrast, recent works in the natural image domain investigated to draw markers directly on the image leading to state-of-the-art results in zero-shot tasks (Shtedritski et al., 2023; Yang et al., 2024). This method is particularly appealing because it requires no additional training or fine-tuning, allowing for post-hoc application even in pretrained models. Moreover, it eliminates the need for extensive datasets with spatial annotations, which are scarce in radiology. While Shtedritski et al. (2023) hypothesize that this emergent capability is limited to models trained on very large datasets, we propose that models exposed to scientific literature, which frequently includes visual markers, may also develop this ability. Therefore, we follow the approach from Shtedritski et al. (2023) and draw the visual prompts directly in the image. We study a range of visual prompts in shape and color, inspired by common highlighting techniques in the medical literature (Appendix Fig. 3). Specifically, we experiment with: arrows, which point at the target object, bounding boxes and circles surrounding the target object. We assume access to the bounding box coordinates for the region of interest. For the bounding box marker, the predefined bounding boxes are directly drawn on the image. The circle marker is represented by an ellipse encompassing the given bounding box coordinates. The arrow marker extends from the image center to the bounding box center, with a length of at least 25% of the smaller image dimension to ensure visibility. If the bounding box center coincides with the image center, a slight offset is applied to avoid a zero-length arrow.

The modified images are then processed by the image encoder and classified using the previously described approach (Sec. 2.1).

**Text Prompts**    The text prompts used in our experiments follow a standardized template: "A chest X-ray with signs of {class}." For binary malignancy classification, we adapt this format to "A chest X-ray with a {malignancy} {class}." where {malignancy} is either "malignant" or "benign".

To investigate the effect of explicitly referencing visual markers, we conduct an ablation study by modifying the prompts to include marker descriptions. Specifically, we append "indicated by a {color} {annotation}." where {annotation} represents the type of marker (arrow, bounding box, or circle) and {color} corresponds to the applied visual marker color.

### 2.3. Evaluation

**Quantitative Evaluation**    We quantitatively evaluate the effect of visual prompts using AUROC for multi-label and binary classification. In the multi-label setting, we macro average the class-wise AUROC (Hanley and McNeil, 1982; Maier-Hein et al., 2024). Since in the multi-label setting, there can be multiple pathologies in a single image, the evaluation without any cropping or prompting is not straightforward, since usually only the text prompt with the highest probability is selected. Therefore, if there is more than one pathology, we choose the top $M$ predicted classes, with $M$ being the number of ground truth pathologies in the image. In cases where we apply visual prompting, we only utilize the highest class probability, since we provide $M$ images with different visual prompts. This approach slightly favors the non-prompting case, since for the prompting case, each prediction is independent, therefore allows multiple times the same prediction, which is not possible in our selected datasets.

**Explainability**  To assess whether visual prompts improve not only classification performance but also the model's ability to focus on relevant regions, we employ LeGrad (Bousselham et al., 2024) as an explainability method. LeGrad computes gradients with respect to the attention maps of the ViT layers, using these gradients as an explainability signal. It has demonstrated superior spatial fidelity and robustness to perturbations compared to other state-of-the-art explainability methods (Bousselham et al., 2024). We qualitatively compare the attention maps of images with and without visual prompts to evaluate whether the model focuses on the intended regions.

## 3. Experiments

### 3.1. Dataset

To evaluate our proposed approach, we utilize four public chest X-ray datasets with location annotations for the pathologies. A more detailed description about the datasets can be found in Appendix A.

**Padchest-GR**  includes 4,555 chest X-ray (CXR) studies with grounded radiology reports and bounding box annotations (Castro et al., 2024). We filter for samples where each pathology has a only single bounding box to ensure fair comparison with our cropping baseline.

**VinDr-CXR**  includes 18,000 chest X-ray (CXR) scans with radiologist-annotated bounding boxes for 22 findings (Nguyen et al., 2022). We use the official train and test split and apply the same filtering criteria to retain only samples with a single bounding box per pathology.

**Chestx-ray8 (NIH14)**  consists of 108,948 frontal-view chest X-ray images labeled with eight common thoracic diseases extracted via natural language processing from radiology reports (Wang et al., 2017). A subset of 983 images includes manually annotated bounding boxes for 1,600 pathology instances, which we use for our study.

**JSRT**  includes 154 chest X-Rays with a lung nodule (100 malignant and 54 benign nodules) including the X and Y coordinates, and the size of the nodule (Shiraishi et al., 2000).

### 3.2. Models

We evaluate our proposed approach on two biomedical vision-language models, BiomedCLIP and BMCA-CLIP, both trained on scientific biomedical image-text pairs from PubMed Central (PMC) using the CLIP framework. For both models, we use the official HuggingFace (Wolf et al., 2019) models and implementation, including the preprocessing.

**BiomedCLIP**  is pretrained on 15 million PMC-derived image-text pairs and adapts CLIP for biomedical tasks, using PubMedBERT as the text encoder and an ImageNet-pretrained ViT-B/16 as the image encoder. It has demonstrated state-of-the-art performance in image classification, retrieval, and visual question answering (VQA), even outperforming some radiology-specific models on chest X-ray benchmarks (Zhang et al., 2023a).

Table 1: Zero-shot classification performance (AUROC) of BiomedCLIP and BMCA-CLIP on four chest X-ray datasets. Across most datasets, visual prompt markers improve the classification performance. In most cases, mentioning the marker in prompt further improves the performance. Colors are normalized by model and column.

| | Visual Prompt | Marker in text prompt | Padchest-GR | | | VinDr-CXR | | NIH14 | JSRT |
|---|---|---|---|---|---|---|---|---|---|
| | | | Train | Val | Test | Train | Test | | |
| **BiomedCLIP** | No visual prompt | | 0.607 | 0.633 | 0.621 | 0.612 | 0.629 | 0.705 | 0.550 |
| | Crop | | 0.751 | 0.715 | 0.744 | 0.659 | 0.693 | 0.758 | 0.508 |
| | Arrow | | 0.722 | 0.713 | 0.726 | 0.717 | 0.736 | 0.734 | 0.548 |
| | Arrow | ✓ | 0.724 | 0.723 | 0.731 | 0.732 | 0.745 | 0.735 | 0.541 |
| | BBox | | 0.737 | 0.772 | 0.768 | 0.687 | 0.722 | 0.760 | 0.519 |
| | BBox | ✓ | 0.761 | 0.803 | 0.784 | 0.698 | 0.730 | 0.775 | 0.543 |
| | Circle | | 0.753 | 0.784 | 0.775 | 0.683 | 0.727 | 0.762 | 0.555 |
| | Circle | ✓ | 0.771 | 0.799 | 0.791 | 0.675 | 0.725 | 0.773 | 0.568 |
| **BMCA-CLIP** | No visual prompt | | 0.582 | 0.613 | 0.604 | 0.526 | 0.589 | 0.624 | 0.484 |
| | Crop | | 0.706 | 0.692 | 0.728 | 0.577 | 0.606 | 0.701 | 0.548 |
| | Arrow | | 0.688 | 0.698 | 0.709 | 0.617 | 0.648 | 0.640 | 0.517 |
| | Arrow | ✓ | 0.691 | 0.706 | 0.711 | 0.618 | 0.651 | 0.638 | 0.525 |
| | BBox | | 0.757 | 0.791 | 0.789 | 0.599 | 0.660 | 0.668 | 0.502 |
| | BBox | ✓ | 0.752 | 0.788 | 0.777 | 0.598 | 0.653 | 0.665 | 0.461 |
| | Circle | | 0.763 | 0.781 | 0.783 | 0.614 | 0.660 | 0.678 | 0.505 |
| | Circle | ✓ | 0.766 | 0.788 | 0.780 | 0.624 | 0.665 | 0.679 | 0.503 |

**BMCA-CLIP** is trained on 24 million image-text pairs from BIOMEDICA, extends this approach with continual pretraining and streaming-based optimization, using a ViT-L/14 image encoder and PubMedBERT text encoder. It achieves state-of-the-art zero-shot classification across 40 biomedical tasks while requiring $10\times$ less compute than previous models (Lozano et al., 2025).

### 3.3. Visual Prompting Details

For all datasets, we utilize the provided location information of pathologies to generate visual prompts. These prompts are based on bounding box coordinates, which highlight the regions of interest within the images. For the JSRT dataset, nodule locations are given as center coordinates (X, Y) along with the nodule size in mm. To define a bounding box, we first convert the nodule size from millimeters to pixels based on the dataset resolution. A square bounding box is then centered at (X, Y), with its side length equal to the converted nodule size in pixels. For all other datasets, bounding box coordinates are directly provided. As a baseline, we evaluate cropping, where the image is cropped around the bounding box centroid while maintaining a minimum crop size. The cropping dimensions are dynamically adjusted to be at least the bounding box size or 20% of the image size, ensuring a balance

between focus on the pathology and preserving contextual information. To prevent out-of-bounds errors, the final crop is constrained within the image boundaries.

Through preliminary experiments, we find that a red line with a width of 1 achieves the highest average performance across models and visual markers (Appendix Table 2 and Table 3). Therefore, all experiments are conducted using this configuration.

## 4. Results

### 4.1. Quantitative Results

Our results (Table 1) show that focusing the model on the region of interest, either via cropping or visual markers, consistently improves zero-shot classification performance compared to no visual prompt across all datasets. This confirms that guiding attention to pathology regions enhances the model's discriminative capabilities. Among visual markers techniques, bounding boxes and circles emerge as the most robust choices across datasets, with circle prompts performing particularly well. Cropping remains competitive but is generally outperformed by visual markers, except for BMCA-CLIP on NIH14 and JSRT, where cropping achieves the highest performance. For BiomedCLIP, visual markers consistently outperform cropping on all datasets, with bounding boxes and circles leading on PadChest-GR, and arrows performing best on VinDR-CXR.

While the JSRT dataset is only binary, the task of differentiating between malignant and benign nodules is notoriously challenging even for experienced radiologists (MacMahon et al., 2017). This inherent difficulty likely underlies the uniformly low performance scores observed across all approaches. Notably, however, the circle marker in BiomedCLIP achieves the highest performance.

As shown in Appendix Fig. 5, small pathologies particularly benefit from visual prompting. This is likely because, without explicit markers, smaller lesions are more prone to being overlooked by the models. Visual prompts therefore appear especially beneficial when pathology visibility is low, further underscoring their relevance in radiological applications.

Additionally, incorporating visual marker descriptions into text prompts further enhances performance in most cases, indicating a synergistic effect between textual and visual cues. This suggests that explicitly referencing visual prompts in text helps align the model's attention with the pathology region.

Since visual markers directly modify the input images, they inherently carry the risk of occluding diagnostically relevant features. Additionally, their effectiveness could depend on precise marker placement and size. To assess the robustness of visual prompting under realistic variations, we conduct ablation studies evaluating sensitivity to spatial shifts and marker scaling.

In Appendix Fig. 4, panel (a) quantifies the effect of shifting the marker up to 25 % away from the ground truth location in randomly chosen directions. This simulates realistic localization uncertainty. Despite a gradual performance decline with increasing displacement, visual prompting consistently outperforms the no-visual-prompt baseline across shifts. This highlights the approach's robustness to moderate localization errors.

To further evaluate robustness, we assess sensitivity to changes in marker size (Appendix Fig. 4, panel (b)). Specifically, we shrink and enlarge the markers by up to 25 % relative

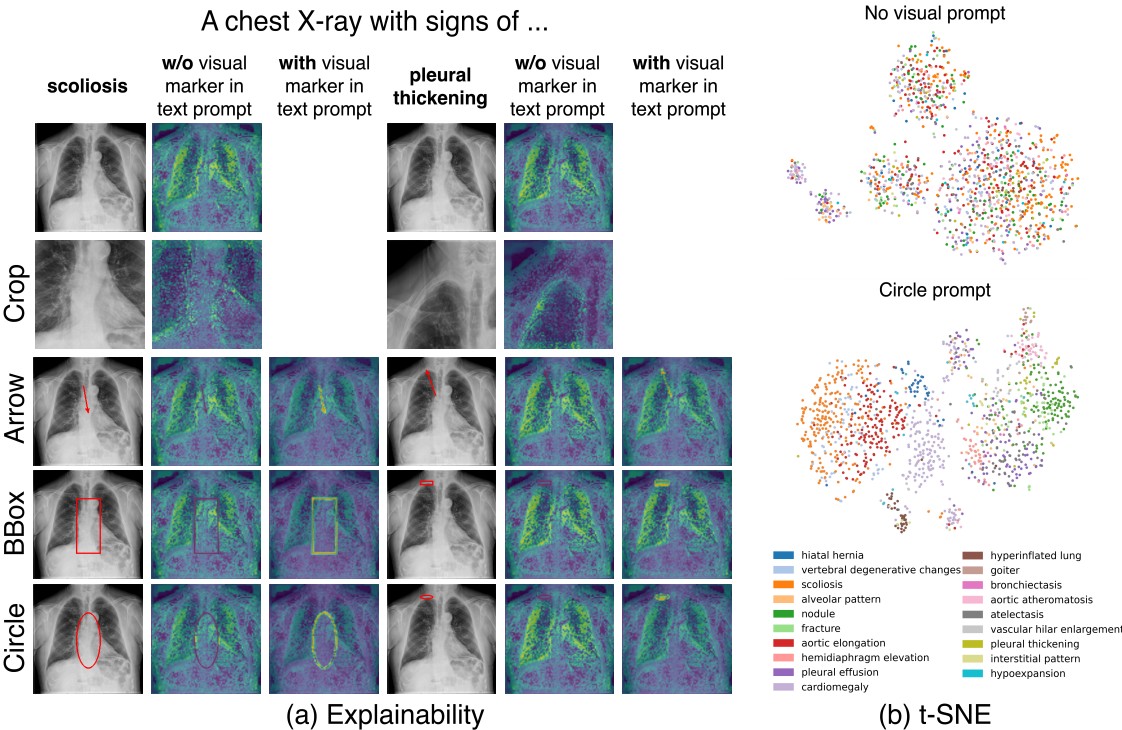

(a) Explainability          (b) t-SNE

Figure 2: (a) Input images and LeGrad attention maps for BMCA-CLIP with different visual prompts. Each row corresponds to a distinct visual prompt. The first and fourth columns display the input images, while the remaining columns show LeGrad attention maps. The second and fifth columns depict attention maps when no visual marker description was included in the text prompt, whereas the third and sixth columns show attention maps when the visual marker was explicitly mentioned. (b) t-SNE projection of single-class samples from the PadChest-GR dataset, with pathologies color-coded. The top plot represents BMCA-CLIP's image embeddings without visual prompts, while the bottom plot shows embeddings with a red circle prompt. The addition of visual prompts enhances clustering, suggesting improved model focus on pathology-relevant features.

to the original ground truth bounding box. While performance varies with marker size, all visual prompting conditions substantially outperform the no-prompting baseline.

In most cases, shrinking the marker reduces performance, likely due to insufficient visibility of the diagnostically relevant regions. Interestingly, for many configurations, enlarging the marker beyond the ground truth region actually improves performance, suggesting that slightly expanding the highlighted area can enhance the model's ability to detect the pathology.

### 4.2. Qualitative Results

**Explainability**   To better understand the impact of visual prompts, we employ LeGrad (Bousselham et al., 2024), an explainability method that visualizes model attention. When visual markers are mentioned in the text prompt, the model demonstrates increased focus on the relevant pathology regions, as shown in the attention maps (Fig. 2(a)). This suggests that visual prompts not only improve classification performance but also enhance model interpretability, ensuring that the model attends to clinically relevant areas.

**t-SNE**   Visual prompt markers alter the input image while refining the model's focus, which should ideally result in more distinct and pathology-aligned feature embeddings. To test this hypothesis, we analyze embedding clusters using t-SNE (Van der Maaten and Hinton, 2008). Specifically, we apply t-SNE to a single-class subset of PadChest-GR to observe whether visual prompting improves the clustering of pathology representations.

As shown in Fig. 2 (b), pathology clusters appear more distinct and well-separated when using a circle visual prompt, compared to no visual prompt. This indicates that visual prompting enhances feature representation, making embeddings more discriminative and aligned with pathology characteristics.

## 5. Conclusion

This study demonstrates that incorporating visual cues can significantly enhance the zero-shot classification performance of Vision-Language Models (VLMs) for radiological images. By leveraging visual markers such as arrows, bounding boxes, and circles, alongside corresponding text prompts, we observed consistent performance improvements across multiple public datasets. Beyond improving classification accuracy, our results show that visual cues help guide model attention to clinically relevant areas, as evidenced by attention maps and feature clustering analyses.

Importantly, our work goes beyond visual prompt engineering by exploring how spatial information can improve zero-shot localized classification. To support further research, we rely exclusively on public datasets and release our code and preprocessing pipeline, allowing for standardized benchmarking in localized classification for medical imaging. We hope this serves as a useful reference for future work and contributes to improving the integration of visual cues in zero-shot medical image classification.

## Acknowledgments

We thank Piotr Kalinowski, Julius Holzschuh and Paul F. Jäger. This study was partially funded by NUM 2.0 (FKZ: 01KX2121).

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

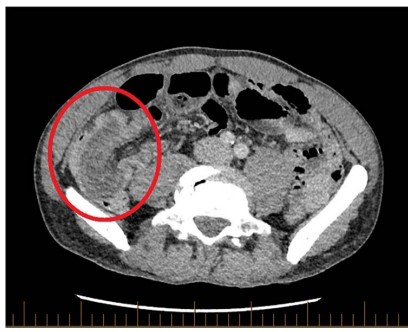 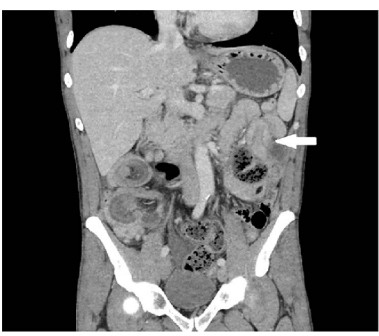 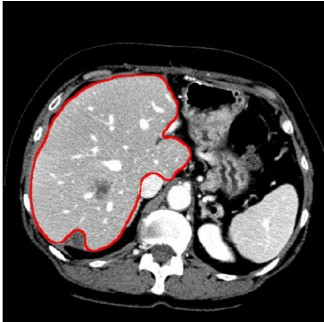

Axial CT of the abdomen and pelvis with portal venous contrast. Long segment of ileocolic intussusception shown (red circle).

Coronal CT for abdomen and pelvis with portal venous. Jejunal intussusception in left upper quadrant (white arrow).

Contour of the liver segmentation (in red) compared to the original CT scan slice

Figure 3: Example figures from PubMedCentral (Turco, 2024; Massoptier and Casciaro, 2008) containing visual markers to guide the reader on specific regions of interest. Those markers are also referred to in the figure descriptions.

## Appendix A. Datasets

**Padchest-GR** includes 4,555 chest X-ray (CXR) studies with grounded radiology reports and bounding box annotations (Castro et al., 2024). We filter for samples where each pathology has a only single bounding box to ensure fair comparison with our cropping baseline. We use the official train, validation, and test split but filter for samples where each pathology has a only single bounding box to ensure fair comparison with our cropping. For the training set, this results in 1,547 images with a total of 19 classes: hiatal hernia, vascular hilar enlargement, atelectasis, cardiomegaly, nodule, aortic atheromatosis, aortic elongation, scoliosis, vertebral degenerative changes, alveolar pattern, hypoexpansion, pleural effusion, hemidiaphragm elevation, fracture, pleural thickening, hyperinflated lung, goiter, bronchiectasis, interstitial pattern. For the validation set, this results in 221 samples and 20 classes, with the same classes as the training set, plus osteopenia. For the test set, this results in 446 samples and 19 classes, with the same classes as the training set, except osteopenia missing.

**VinDr-CXR** includes 18,000 chest X-ray (CXR) scans with radiologist-annotated bounding boxes for 22 findings (Nguyen et al., 2022). We use the official train and test split and limit our selection to samples where each pathology has only a single bounding box, ensuring fair comparison with our cropping baseline. Furthermore, we exclude samples labeled as 'Other lesion' due to their lack of specificity. For the training set, this results in 2602 images with in total 21 classes: Infiltration, Lung Opacity, Consolidation, Nodule/Mass, Aortic enlargement, Cardiomegaly, Pleural effusion, Pulmonary fibrosis, Pleural thickening, Enlarged PA, ILD, Lung cavity, Atelectasis, Calcification, Mediastinal shift, Clavicle fracture, Pneumothorax, Rib fracture, Emphysema, Lung cyst, Edema. For the test set, this results in 609 samples and the same classes, except Emphysema, Lung cyst and Edema missing.

Table 2: Ablation study on line width of visual prompt marker. We fix the marker color to red and evaluate AUROC on the PadChest-GR test set, averaging results across conditions where the marker was and was not mentioned in the text prompt.

|  | 1 | 2 | 3 | 4 | 5 | 7 | 10 |
|---|---|---|---|---|---|---|---|
| BiomedCLIP |  |  |  |  |  |  |  |
| Arrow | **0.728** | 0.718 | 0.712 | 0.709 | 0.699 | 0.689 | 0.685 |
| BBox | **0.776** | 0.770 | 0.761 | 0.756 | 0.751 | 0.736 | 0.720 |
| Circle | **0.783** | 0.775 | 0.770 | 0.764 | 0.758 | 0.746 | 0.741 |
| BMCA-CLIP |  |  |  |  |  |  |  |
| Arrow | **0.710** | 0.706 | 0.705 | 0.702 | 0.697 | 0.691 | 0.679 |
| BBox | **0.783** | 0.778 | 0.772 | 0.764 | 0.755 | 0.741 | 0.717 |
| Circle | **0.782** | 0.765 | 0.761 | 0.755 | 0.752 | 0.742 | 0.714 |
| Average | **0.760** | 0.752 | 0.747 | 0.742 | 0.735 | 0.724 | 0.709 |

Table 3: Ablation study on color of the visual prompt marker. We fix the line width to 1 and evaluate AUROC on the PadChest-GR test set, averaging results across conditions where the marker was and was not mentioned in the text prompt.

|  | Black | Blue | Green | Orange | Red | White | Yellow |
|---|---|---|---|---|---|---|---|
| BiomedCLIP |  |  |  |  |  |  |  |
| Arrow | **0.730** | 0.718 | 0.712 | 0.709 | 0.728 | 0.723 | 0.708 |
| BBox | 0.760 | 0.753 | 0.775 | 0.765 | **0.776** | 0.745 | 0.759 |
| Circle | 0.772 | 0.780 | **0.794** | 0.779 | 0.783 | 0.752 | 0.773 |
| BMC-CLIP |  |  |  |  |  |  |  |
| Arrow | 0.691 | 0.704 | 0.694 | **0.713** | 0.710 | 0.696 | **0.713** |
| BBox | 0.757 | 0.776 | 0.768 | 0.769 | **0.783** | 0.743 | 0.768 |
| Circle | 0.765 | **0.783** | 0.779 | 0.782 | 0.782 | 0.765 | 0.780 |
| Average | 0.746 | 0.752 | 0.754 | 0.753 | **0.760** | 0.737 | 0.750 |

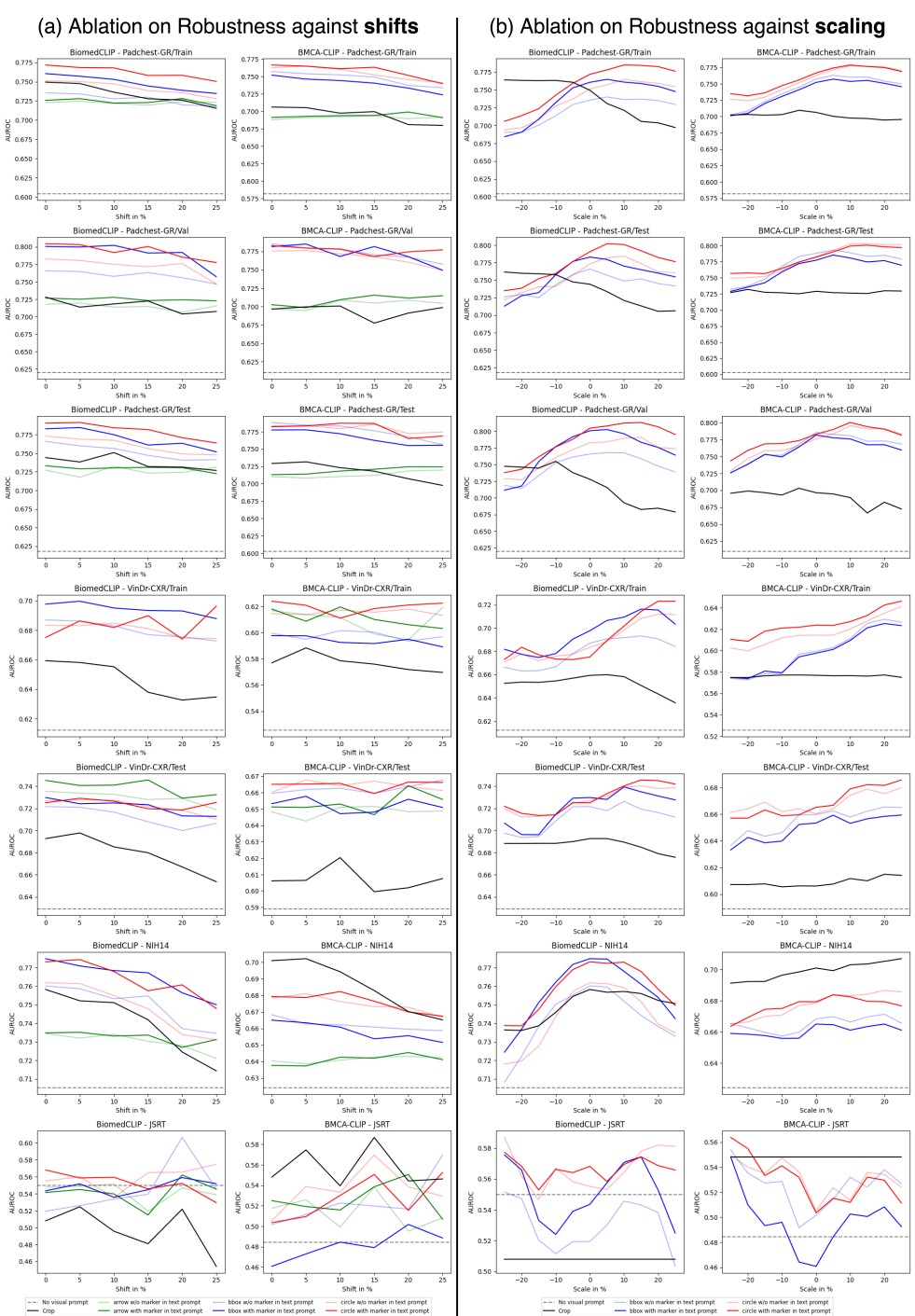

Figure 4: Ablation study on robustness. (a) Evaluation of robustness to spatial shifts of the visual prompt across models and datasets. (b) Performance assessment across different scales of the visual prompt marker.

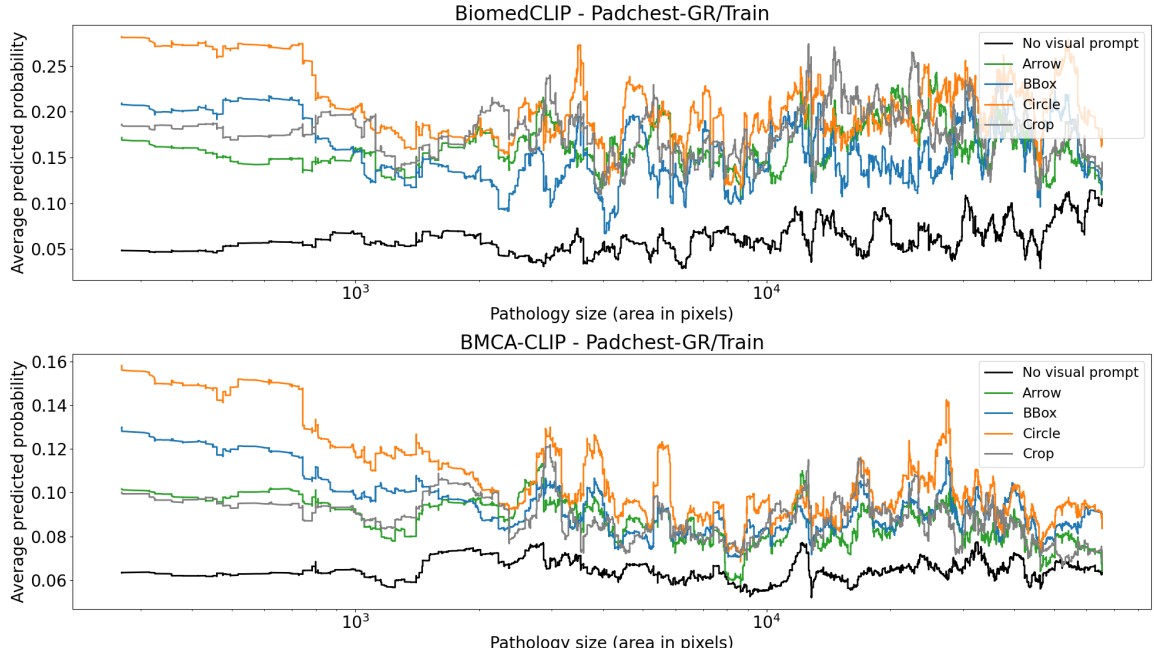

Figure 5: Average predicted probability (y-axis) as a function of pathology size (x-axis) for the PadChest-GR training dataset. For each image, we extract the softmax probability assigned to the ground truth class and compute a moving average across pathology sizes. Results are shown for both models. BiomedCLIP (top) and BMCA-CLIP (bottom). Particularly, small pathologies benefit from visual prompting.

