# OpenReview forum: "Visual Prompt Engineering for Vision Language Models in Radiology"
_MIDL.io/2025/Conference — MIDL 2025 Poster_

### Official Review · Reviewer_mtRp · 2025-02-21

**Confidence:** 4
**Preliminary Rating:** 2
**Recommendation:** Poster
**Final Rating:** 4

**Summary:**

This paper investigates the application of visual prompts in a medical CLIP framework.  Through classification experiments, the authors demonstrate that an appropriately designed visual prompt can enhance the model's classification accuracy.  Moreover, the study includes visualizations that elucidate how the model processes the visual prompt.

**Strengths:**

- The topic is engaging and contributes significantly to understanding the alignment of encoded features in multimodal models.

- Dataset selection is appropriate for the research objectives.

- The manuscript is well-written and easily comprehensible.

**Weaknesses:**

- **Repetition of Prior Work:** The study largely replicates the findings of [1] and reaches similar conclusions, merely indicating that a red circle marking is the optimal choice for enhancing accuracy.
- **Novelty Claim:** The manuscript's claim of being the first to explore visual prompts in radiology is inaccurate. The medical adaptation of SAM (MedSAM) [3], based on SAM [2], already employs marking techniques (e.g., points, bounding boxes) to highlight regions of interest and improve segmentation accuracy.
- **Experimental Focus:** The experiments are limited to zero-shot classification accuracy and do not demonstrate how visual prompts might benefit other tasks, such as radiology report generation, especially given that similar conclusions have been reported in [1].
- **Overlooking Details:** In medical imaging, lesions are often very small. The direct overlay approach used in the study may inadvertently cause both clinicians and models to overlook important diagnostic details.
- **Unpractical Data Cleansing Criteria:** The strict criterion of allowing only one occurrence of each disease-specific phenomenon is excessively rigid and impractical for real-world applications.


[1]. Shtedritski, Aleksandar, Christian Rupprecht, and Andrea Vedaldi. "What does clip know about a red circle? visual prompt engineering for vlms." Proceedings of the IEEE/CVF International Conference on Computer Vision. 2023.

[2]. Kirillov, Alexander, et al. "Segment anything." Proceedings of the IEEE/CVF international conference on computer vision. 2023.

[3]. Ma, Jun, et al. "Segment anything in medical images." Nature Communications 15.1 (2024): 654.

**Detailed Comments:**

- **Model Selection:** For classification tasks under limited data conditions, consider employing a smaller model, which might be more practical and efficient.

- **Enhancing Visual Prompt Encoding:** Investigate the incorporation of additional layers or alternative model architectures for encoding visual prompts. This may improve both the interpretability and flexibility of the approach.

- **Extended Application:** Explore deeper applications of visual prompts, such as guiding attention to specific regions for anomaly detection or generating X-ray chest images that highlight designated lesions.

**Justification Of The Final Rating:**

The VLM has become more and more important in the medical field. The main reason that change my mind is the extra experiment that shows the prompt can also works alone for the 'old' task(classification). I would think this paper will make more people works in medical field will also appreciate that the tricks and prompts which help in the natural image can also help with medical image.

**Justification Of The Preliminary Rating:**

All in all, while the paper addresses an interesting topic, it suffers from limited novelty given its close resemblance to prior work, and the experimental focus that restricts its broader applicability, and the data-cleansing criteria that may not translate well to clinical settings.

Additionally, the lack of comparison with simpler baselines, such as a fine-tuned pre-trained ViT model with an MLP layer makes me worry about the necessity. I would suggest the author either consider how to improve the classification performance by the prompt or add additional tasks to show the necessity of VLM.

**Questions To Address In The Rebuttal:**

I observed that the results on the JSRT dataset appear to be completely unsuccessful, despite the task being a binary classification. Could the authors provide insight into the underlying reasons for this failure?

Have the authors considered whether simply fine-tuning a pre-trained ViT model with an MLP layer for the classification task would similarly benefit from the use of visual prompts?

---

> ### Author Response · Authors · 2025-03-08
>
> Thank you again for your valuable comments.
>
> W1. We thank the reviewer for raising the concern of repetition of prior work.
> - While visual prompt engineering has been explored in other domains, our work is the first to investigate it in the radiological domain. Transferring promising concepts from general vision-language modeling (such as CLIP[4] or SAM[2]) into specialized domains such as medical imaging has consistently demonstrated significant scientific value, as shown by recent high-impact studies (e.g., MedCLIP[5], MedSAM[3]).
> - Additionally, Shtedritski et al. [1] hypothesized that the ability to recognize and utilize visual markers is primarily learned from large datasets and high-capacity models due to the scarcity of such markers in training data. Our work challenges this assumption by demonstrating that VLMs trained on scientific literature—despite being trained on significantly smaller datasets (15M/24M vs. 400M images)—also exhibit this capability. This distinction is critical, as scientific literature frequently employs visual markers to highlight pathologies, making our approach particularly relevant to medical imaging.
> - Beyond validating this hypothesis, our study also contributes by establishing the first benchmark for region-guided classification in radiology, providing a foundation for future methodological advancements in this area.
> - Thus, rather than merely replicating prior work, our study extends the understanding of visual prompts in a new domain, challenges existing assumptions, and lays the groundwork for further research in radiological applications.
>
> W2. We sincerely appreciate the reviewer’s careful evaluation of our novelty claim
> - In our manuscript, we define visual prompt engineering as the embedding of markers directly within the radiological image. We state that “[…] this work investigates whether visual prompt engineering—embedding markers within radiological images—enhances zero-shot classification performance. […] To our knowledge, this is the first study to investigate visual prompt engineering in the radiological domain.”
> - Our approach is distinct from methods such as SAM[2] and MedSAM[4], which employ dedicated prompt encoders that require additional training on task-specific data. In contrast, we leverage pre-trained off-the-shelf vision-language models (VLMs) and modify only the input image, thus avoiding dedicated training.
> - We have added a clarifying sentence to the manuscript to explicitly distinguish our method from those approaches.
>
> W3. We appreciate the reviewer’s feedback on the experimental focus and extended application
> - We agree that applications beyond zero-shot classification—such as radiology report generation or image-to-image generation—are of great interest. However, given that CLIP-based VLMs are encoder-only architectures, extending our approach to these tasks would necessitate additional modifications and training.
> - We consider this an important direction for future work.
>
> W4. We appreciate the reviewer’s insightful comment regarding the potential risk of overlooking small lesions due to the direct overlay approach.
> - We acknowledge that directly overlaying markers may risk overlooking subtle diagnostic details. Although prior works (e.g., MedSAM [5], SAM[2], Alpha-clip[6]) integrate region-specific prompting within their architectures through dedicated training on precisely annotated data, such approaches are often costly and infeasible in medical imaging.
> - Importantly, our experiments show that overall performance improves with the use of visual markers.
> - To investigate potential losses in diagnostic detail, we conducted an extensive new ablation study on scaling the bounding box.
> - The results (new Appendix Figure 4) indicate that in most cases, enlarging the bounding box beyond the ground truth improves performance—likely compensating for any loss in visibility of critical features.
> - These findings have been added to the results section.
>
> W5. We acknowledge that our data cleansing criteria may seem overly strict for real-world applications.
> - However, to ensure a fair evaluation, we apply the same data cleansing criteria across all methods.
> - While our approach is capable of handling multiple occurrences of the same pathology, the no-visual-prompt baseline is inherently limited to classifying each pathology only once.
> - Constraint to only a single occurrence per pathology ensures that differences in performance are attributable to the method itself rather than inconsistencies in data preprocessing.
>
> [4] Radford, Alec, et al. "Learning transferable visual models from natural language supervision." International conference on machine learning. PmLR, 2021.
>
> [5] Ma, Jun, et al. "Segment anything in medical images." Nature Communications 15.1 (2024): 654.
>
> [6] Sun, Zeyi, et al. "Alpha-clip: A clip model focusing on wherever you want." Proceedings of the IEEE/CVF conference on computer vision and pattern recognition. 2024.

---

> > ### Author Response · Authors · 2025-03-08
> >
> > Q1. We appreciate the reviewer’s observation regarding the performance on the JSRT dataset.
> > - Distinguishing malignant from benign pulmonary nodules on X-rays is inherently challenging due to the two-dimensional nature of radiographs, overlapping anatomical structures, and limited resolution that hinders the characterization of subtle features critical for malignancy assessment [7].
> > - Consequently, it is unsurprising that zero-shot approaches exhibit uniformly low performance on this task.
> > - Notably, however, the circle marker in BiomedCLIP achieves the highest performance, indicating that our approach still confers benefits in this challenging setting.
> >
> > Q2. We appreciate the reviewer’s concern regarding the lack of comparison with simpler baselines.
> > - To address this, we conducted a fine-tuning experiment on the PadChest-GR dataset (using its predefined train, validation, and test splits).
> > - Fine-tuning was performed in two stages:
> >      - Training only the linear head for 10 epochs (learning rate: 1e-4).
> >      - Unfreezing the backbone (BiomedCLIP or BMCA-CLIP vision encoder) and training the full network for 100 epochs (learning rate: 1e-6, batch size: 32) with early stopping based on validation loss (cross-entropy, patience: 10 epochs).
> > - Our results demonstrate that fine-tuning with visual prompts improves AUROC compared to training without them.
> > - Also the performance is substantially higher for the fine-tuning model over the zero-shot visual prompting model.
> > - However, this improvement requires labeled data and computational resources.
> > - Moreover, a fine-tuned classifier is limited to the training labels, whereas zero-shot classification inherently supports open-set recognition.
> > - While fine-tuning can enhance performance in fixed-label scenarios, our focus on zero-shot classification is driven by its flexibility and scalability.
> >
> > BiomedCLIP results:
> > |                   | Visual Prompting | Finetuning |
> > |-------------------|-----------------|------------|
> > | No visual prompt | 0.621           | 0.684      |
> > | Crop             | 0.744           | 0.944      |
> > | Arrow            | 0.731           | 0.868      |
> > | BBox             | 0.784           | 0.954      |
> > | Circle           | 0.791           | 0.952      |
> >
> > BMCA-CLIP  results:
> > |                   | Visual Prompting | Finetuning |
> > |-------------------|-----------------|------------|
> > | No visual prompt | 0.604           | 0.698      |
> > | Crop             | 0.728           | 0.955      |
> > | Arrow            | 0.711           | 0.916      |
> > | BBox             | 0.777           | 0.964      |
> > | Circle           | 0.780           | 0.968      |
> >
> >
> > [7] MacMahon, Heber, et al. "Guidelines for management of incidental pulmonary nodules detected on CT images: from the Fleischner Society 2017." Radiology 284.1 (2017): 228-243.

---

> > > ### Comment · Reviewer_mtRp · 2025-03-08
> > >
> > > Thank you so much for your detailed reply, that’s answered all my questions. I would like to improve my score to WA later.

---

> > > > ### Author Response · Authors · 2025-03-08
> > > >
> > > > We sincerely appreciate the reviewer’s thoughtful engagement with our work and their willingness to reconsider their score. We are glad that our responses addressed all concerns, and we thank the reviewer for their valuable feedback, which has helped improve the clarity and rigor of our paper.

---

> > > > ### Comment · Reviewer_mtRp · 2025-03-08
> > > > **Reason to increase score**
> > > >
> > > > - Interesting experiment
> > > > - Extra experiment during rebuttal shows that vision prompts can help with more general tasks and make unique contributions.

---

### Official Review · Reviewer_vE7P · 2025-02-27

**Confidence:** 3
**Preliminary Rating:** 3

**Summary:**

Capturing local image content to improve interpretability is needed in radiology and is not addressed by CLIP. The authors propose to use visual markers (arrows, bounding boxes, etc) in radiological images to guide model attention. Experimental results show that the proposed method improves AUROC.

**Strengths:**

The experiments are conducted on 4 public datasets and two biomedical vision-language models are used. Explainability results are included. Ablations on line width of visual prompt marker and on color of the visual prompt marker are provided in the appendix.

**Weaknesses:**

Given that the visual markers play an important role, how the visual markers are placed may greatly affect the performance of the proposed approach. In the experiments, the authors use bounding boxes centered at the nodule locations. The nodule locations are annotated by radiologists and this limits the practicality of the proposed approach.

**Detailed Comments:**

It would be nice if the authors can provide ablation results demonstrating that the proposed method is robust against slight changes in the marker placement and thus showcasing that annotations done by non-experts can also help.

**Justification Of The Preliminary Rating:**

The experimental settings are comprehensive. However, the lack of ablation results over the bounding box / marker locations limits the practicality of the proposed method. I am willing to increase the score if the authors can provide these ablations.

**Questions To Address In The Rebuttal:**

Please see detailed comments above.

---

> ### Author Response · Authors · 2025-03-08
>
> Thank you again for your valuable comments.
>
> We appreciate the reviewer’s valuable suggestion on evaluating the effect of marker size and positioning. In response, we conducted extensive new experiments, covering over 1,300 hyperparameter configurations, to assess the robustness of our approach against shifts and scaling of the visual markers.
> - In the new Appendix Figure 4, panel (a) we quantify the effect of shifting the marker up to 25 % away from the ground truth location in randomly chosen directions. This experiment simulates realistic localization uncertainty that might arise when using non-expert annotations.
> - Despite a gradual performance decline with increasing displacement, visual prompting consistently outperforms the no-visual-prompt baseline across all shifts. This highlights the approach’s robustness to moderate localization errors.
> - To further evaluate the practical feasibility of using non-expert annotations, we also assess sensitivity to marker size variations in Appendix Figure 4, panel (b). We explore marker scaling by up to ±25% relative to the original ground truth bounding box.
> - The results demonstrate that while performance varies with marker size, all visual prompting conditions continue to substantially outperform the no-prompting baseline.
> - Interestingly, in most cases, shrinking the marker reduces performance, likely due to insufficient visibility of the diagnostically relevant regions.
> - However, for many configurations, enlarging the marker beyond the ground truth region actually improves performance, suggesting that slightly expanding the highlighted area can enhance the model’s ability to detect the pathology.
> - These new ablation experiments and discussions have been incorporated into the manuscript.
> - We thank the reviewer for this insightful suggestion, which has helped strengthen the paper.

---

### Official Review · Reviewer_f5ya · 2025-03-01

**Confidence:** 5
**Preliminary Rating:** 5
**Recommendation:** Oral

**Summary:**

This paper proposes how the classification performance of CLIP-based vision-language models (VLMs) in radiological tasks can be enhanced using visual markers. This is an extension of Shtedritski et al.'s work in the natural image domain showing how markers drawn directly on the image lead to state-of-the-art results in zero-shot tasks. The authors performed experiments on public chest X-ray datasets and demonstrated that including visual markers and mentioning them in the text prompt produce higher AUROC values in general across different chest X-ray datasets, along with an analysis on the type of markers. The proposed method also enhances the interpretability of VLMs in terms of attention maps focusing on the region of interest for clinical diagnosis.

**Strengths:**

The paper is well-structured, with a clear outline of the motivation for visual markers in VLMs for disease classification tasks on radiological data. This work utilizes the fact that visual markers are highly prevalent in scientific images, which are also often used for training VLMs in radiology due to limited data availability, and investigated if the presence of such markers indeed improves classification performance. The results indicate that using visual cues in images or text prompts when using VLMs for zero-shot classification are likely to be more useful than directly providing images and text prompts as input without additional markers.

**Weaknesses:**

A potential drawback of using visual marker-based prompts is that it may bias the VLM if there are multiple regions of interest, but many of them are missed by the annotator. An analysis of such potential biases using additional numerical studies may be useful.

**Detailed Comments:**

In Fig. 2a, columns 2 and 5 should be same according to the caption but their titles are different. Same for columns 3 and 6.

**Justification Of The Preliminary Rating:**

Please refer to the Strengths section. Overall, the paper is well-motivated and provides a practical solution to improving prompting in VLMs for disease classification tasks, which are well-supported with relevant experiments along with qualitative and quantitative results.

**Questions To Address In The Rebuttal:**

Please see the section on Weaknesses.

**Special Issue:**

Yes

---

> ### Author Response · Authors · 2025-03-08
>
> Thank you again for your valuable comments and the positive evaluation.
>
> W1: We thank the reviewer for the insightful comment on potential biases.
> - Our approach applies one marker per pathology instance. In images with multiple pathologies, each region of interest is processed independently, with each marker generating a dedicated classification.
> - We acknowledge, that classifying pathologies which are missed by the radiologists is a general limitation of employing visual prompts.
> - If annotations are missed, the setup falls back classifying the unmodified image.
> - Please note, we conducted several new bias-adressing additive experiments regarding shift and scaling of the markers, which showed the robustness of the method (new Appendix Figure 4).
>
> Detailed Comments: We appreciate the reviewer for spotting the mistake in Fig. 2a, which has been corrected accordingly.

---

### Official Review · Reviewer_1qiJ · 2025-03-01

**Confidence:** 3
**Preliminary Rating:** 2
**Final Rating:** 4

**Summary:**

The paper presents a method by adding visual markers to guide the vision-language model's attention. They aim to answer the question whether embedding markers within radiological images will enhances zero-shot classification performance.

**Strengths:**

This is the first work to discuss the visual prompt engineering in the radiological domain. The visual prompts that incorporates arrow, bounding boxes and circle is good. The paper structure is clear and easy to follow.

**Weaknesses:**

1. This is work has limited innovations by just adding visual markers to the prompt and test out the performance. It maybe new in the radiological domain, but this proposal of the adding a focus area in general, has been studied in the past on other image domains [1], [2], [3].

2. Does the visual prompts have a general size preference for circle and bounding boxes? Why don't the authors perform a ablation study regarding to the size choices in the visual prompting? The table 2 and 3 only compares the linewidth and color (which is trivial in my opinion), which is not informative on getting the optimal sizes for the prompts.

3. What about other shapes? Does the guidance shape matters in the visual prompting?

[1] Shtedritski, A., Rupprecht, C., & Vedaldi, A. (2023). What does CLIP know about a red circle? Visual prompt engineering for VLMs. arXiv [Cs.CV]. Retrieved from http://arxiv.org/abs/2304.06712
[2] Gu, J., Han, Z., Chen, S., Beirami, A., He, B., Zhang, G., … Torr, P. (2023). A Systematic Survey of Prompt Engineering on Vision-Language Foundation Models. arXiv [Cs.CV]. Retrieved from http://arxiv.org/abs/2307.12980
[3] Liu, H., Li, C., Wu, Q., & Lee, Y. J. (2023). Visual Instruction Tuning. arXiv [Cs.CV]. Retrieved from http://arxiv.org/abs/2304.08485

**Detailed Comments:**

Since this is a zero-shot focused method, the tested models and datasets are not sufficient in the experiments. I would suggest the authors to perform more intensive evaluation.

**Justification Of The Final Rating:**

The authors addressed my concerns well. Additional experiments including more ablation study have been uploaded as well. The manuscript included more details and explanation.

Thus, I am updating the final rating.

**Justification Of The Preliminary Rating:**

To summarize, my major concern still lays in the novelty of this paper. Simply adding the visual prompts and improving the performance is not considered novel enough for the application. Thus, the overall innovation part is low.

**Questions To Address In The Rebuttal:**

Please refer to the weakness section.

---

> ### Author Response · Authors · 2025-03-08
>
> Thank you again for your valuable comments.
>
> W1.: We thank the reviewer for raising the concern regarding limited innovations.
> - Our paper specifically targets the validation and application track, where the emphasis lies on applying established methodologies to novel domains or practical scenarios, rather than proposing fundamentally new methods. Indeed, transferring promising concepts from general vision-language modeling (such as CLIP[1] or SAM[2]) into specialized domains such as medical imaging has consistently demonstrated significant scientific value, as shown by recent high-impact studies (e.g., MedCLIP[3], MedSAM[4]).
> - In alignment with this scope, our contribution lies precisely in exploring, evaluating, and establishing the effectiveness of visual prompt engineering for vision-language models specifically within the radiological domain, leveraging the prevalence and semantic meaning of visual markers in scientific literature. Additionally, we establish the first benchmark in radiology for evaluating region-guided classification with zero-shot approaches, which provides a valuable resource for future research.
>
>
> W2.: We appreciate the reviewer’s valuable suggestion on evaluating the effect of marker size and positioning. In response, we conducted extensive new experiments, covering over 1,300 hyperparameter configurations, to assess the robustness of our approach against shifts and scaling of the visual markers.
> - In new Appendix Figure 4, panel (a) we quantify the effect of shifting the marker up to 25 % away from the ground truth location in randomly chosen directions. This simulates realistic localization uncertainty. Despite a gradual performance decline with increasing displacement, visual prompting consistently outperforms the no-visual-prompt baseline across all shifts. This highlights the approach’s robustness to moderate localization errors.
> - To further evaluate robustness, we assess sensitivity to changes in marker size (Appendix Figure 4, panel (b)). Specifically, we shrink and enlarge the markers by 25 % relative to the original ground truth bounding box. While performance varies with marker size, all visual prompting conditions substantially outperform the no-prompting baseline. In most cases, shrinking the marker reduces performance, likely due to insufficient visibility of the diagnostically relevant regions. Interestingly, for many configurations, enlarging the marker beyond the ground truth region actually improves performance, suggesting that slightly expanding the highlighted area can enhance the model’s ability to detect the pathology.
> - Furthermore, we provide a new detailed analysis in Appendix Figure 5, that small pathologies particularly benefit from visual prompting. This is likely because, without explicit markers, smaller lesions are more prone to being overlooked by the models. Visual prompts therefore appear especially beneficial when pathology visibility is low, further underscoring their relevance in radiological applications.
> - These new ablation experiments and discussions have been incorporated into the manuscript.
> - We thank the reviewer for this insightful suggestion, which has helped strengthen the paper.
>
> W3.: We thank the reviewer for their insightful comment regarding the choice of guidance shapes.
> - Our selection of arrows, bounding boxes, and circles was motivated by their prevalence in scientific literature.
> - While other markers, such as contour markers are also common, the datasets we used provide only bounding box coordinates or lesion location with diameter, precluding an analysis using contours.
> - Moreover, our experiments (see Table 1) show that the choice of shape matters: circles and bounding boxes in most cases outperform arrows across both models and datasets.
>
> Detailed Comment: We appreciate the reviewer’s interest in further extending the evaluation.
> - Our experiments already encompass a wide range of pathologies, models, and multiple public datasets, as noted by other reviewers.
> - Nonetheless, we remain open to incorporating additional datasets and models, and we welcome further suggestions to strengthen the evaluation.
>
>
> [1] Radford, Alec, et al. "Learning transferable visual models from natural language supervision." International conference on machine learning. PmLR, 2021.
>
> [2] Kirillov, Alexander, et al. "Segment anything." Proceedings of the IEEE/CVF international conference on computer vision. 2023.
>
> [3] Wang, Zifeng, et al. "Medclip: Contrastive learning from unpaired medical images and text." Proceedings of the Conference on Empirical Methods in Natural Language Processing. Conference on Empirical Methods in Natural Language Processing. Vol. 2022. 2022.
>
> [4] Ma, Jun, et al. "Segment anything in medical images." Nature Communications 15.1 (2024): 654.

---

### Author Rebuttal · Authors · 2025-03-08

**Rebuttal:**

We thank all reviewers for their constructive feedback.

We have thoroughly revised our manuscript to address the provided feedback and uploaded the revision, where we highlight the modified parts in red. Changes include:
- We integrated the feedback from the reviewers directly into the manuscript text.
- We added an ablation study on the robustness of the visual prompts (shifts and scaling), now detailed in Appendix Figure 4.
- We investigated the effect of lesion size on performance improvements, now illustrated in Appendix Figure 5.
- We corrected a typo in Figure 2a.

 Please find our point-by-point answers in the respective reviewer sections.

**Supporting Material:**

/attachment/cd59474a14a786f91715c6a880561f12eb065d71.pdf

---

### Meta-Review · Area_Chair_dy3S · 2025-03-22

**Recommendation:** Accept (Poster)
**Confidence:** 4

**Metareview:**

The paper presents an interesting idea of providing visual prompts in VLMs to improve performance of zero-shot classification. Given the idea, the rest of the technical contribution is pretty straightforward, but I think this is a very positive thing.

The reviewers all appreciated the idea, and, although there was some apprehension early on about novelty, after a rigorous discussion period, these worries were alleviated. The authors performed extensive experiments during the rebuttal period, including ablations and sensitivity analysis, which I do believe overall improves the (rigor of) the paper, and the message being communicated at the conference.

Overall, I agree with the reviewer consensus now that the paper can lead to valuable discussion at the conference, and it should be accepted.